# Acquisition of Colistin Resistance Links Cell Membrane Thickness Alteration with a Point Mutation in the *lpxD* Gene in *Acinetobacter baumannii*

**DOI:** 10.3390/antibiotics9040164

**Published:** 2020-04-06

**Authors:** Neveen M. Saleh, Marwa S. Hesham, Magdy A. Amin, Reham Samir Mohamed

**Affiliations:** 1Department of Microbiology, Division of Basic Medical Science, National Organization for Drug Control and Research (NODCAR), 12553 Giza, Egypt; no_more_tears1986@hotmail.com; 2Department of Microbiology and Immunology, Faculty of Pharmacy, University of Cairo, 11562 Cairo, Egypt; magdy.amin@pharma.cu.edu.eg (M.A.A.); reham.samer@pharma.cu.edu.eg (R.S.M.)

**Keywords:** *Acinetobacter baumannii*, colistin, cross-resistance, *LpxD* mutation, pmrB mutation

## Abstract

*Acinetobacter baumannii* is one of the most common causes of nosocomial infections in intensive care units. Its ability to acquire diverse mechanisms of resistance limits the therapeutic choices for its treatment. This especially concerns colistin, which has been reused recently as a last-resort drug against *A. baumannii*. Here, we explored the impact of gaining colistin resistance on the susceptibility of *A. baumannii* to other antibiotics and linked colistin resistance acquisition to a gene mutation in *A. baumannii*. The susceptibility of 95 *A. baumannii* isolates revealed that 89 isolates were multi-drug resistance (MDR), and nine isolates were resistant to colistin. Subsequently, three isolates, i.e., MS48, MS50, and MS64, exhibited different resistance patterns when colistin resistance was induced and gained resistance to almost all tested antibiotics. Upon TEM examination, morphological alterations were reported for all induced isolates and a colistin-resistant clinical isolate (MS34Col-R) compared to the parental sensitive strains. Finally, genetic alterations in *PmrB* and *LpxACD* were assessed, and a point mutation in *LpxD* was identified in the MS64Col-R and MS34Col-R mutants, corresponding to Lys117Glu substitution in the lipid-binding domain. Our findings shed light on the implications of using colistin in the treatment of *A. baumannii*, especially at sub-minimum inhibitory concentrations concentrations, since cross-resistance to other classes of antibiotics may emerge, beside the rapid acquisition of resistance against colistin itself due to distinct genetic events.

## 1. Introduction

*Acinetobacter baumannii* is a highly versatile nosocomial Gram-negative pathogen that has become a critical challenge for common antibiotic treatments [1,2]. Moreover, since it causes catheter-correlated infections, urinary tract infections, pneumonia, wound infections, and bacteremia, its infection is associated with a great risk of hospital death [3].

Carbapenem-resistant *A. baumannii* (crab) was often recorded in hospitals, which forced physicians to reuse the old polymyxin antibiotic, colistin, regardless of its high systematic toxicity, owing to its favorable properties of rapid bacterial killing, a narrow spectrum of activity, and slow development of resistance [1,4]. Unfortunately, colistin resistance has been reported recently [1].

Colistin, a cationic peptide that disrupts Outer membrane(OM), was recognized as a last-resort treatment. Unfortunately, the abuse of colistin, as a consequence of prolonged exposure, is leading to colistin resistance, which is reported periodically [5,6,7]. 

*A. baumannii* showed the heteroresistance phenomenon to colistin, a natural evolution of drug resistance indicating growing colistin resistance in *A. baumannii*, which has therapeutics implications. Sub-populations showing a higher level of resistance to this antibiotic could grow at concentrations four-fold or higher than the MIC of the entire population [1,8,9,10]. 

Moffatt and co-workers attributed the colistin heteroresistance in *A. baumannii* to a loss of lipopolysaccharide production in subpopulations displaying high levels of colistin resistance that were grown by serial passages on colistin plates at increasing colistin concentrations [10]. Cross-resistance reports with different outcomes on the correlation between colistin resistance and resistance to other classes of antibiotics have been reported [11,12]. 

Colistin-resistant *A. baumannii* isolates usually employ several strategies for protection against colistin, among which the modification of their lipopolysaccharides (LPSs) mediated by the two-component system pmrA/pmrB, which have overall negative charges and are the initial targets of colistin [10]. LPSs modifications can involve the addition of phosphoethanolamine (PEtN) and 4-amino-4-deoxy-L-arabinose (L-Ara4N), which change the net negative charge of lipid A to a positive charge, lowering the affinity of LPS for colistin [13,14,15,16], or deacylation and hydroxylation of LPS, but these last mechanisms are less common in *A. baumannii A. baumannii* [4].

In this context, it is also known that the two-component response regulator and sensor kinase *PmrA/B* allows the bacteria to sense and respond to various environmental conditions, including pH and Fe^3+^ and Mg^2+^ levels, and also cause mutations in genes implicated in lipid A modification, thereby influencing the susceptibility to colistin [17,18]. Several studies reported that one or two amino acid substitutions, variously located in *PmrB*, lead to colistin resistance acquisition [18,19]. 

Similarly, the complete loss of the LPS due to impaired lipid A synthesis through mutations in the *lpxA*, *lpxC*, and *lpxD* genes involved in lipid A biosynthesis is another common colistin resistance mechanism. In addition, colistin resistance in *A. baumannii* causes alterations in cell topography, particularly a more spherical appearance, and increased cell surface roughness, leading to outer membrane damage [20,21].

Based on what has been mentioned, we explored the impact of gaining colistin resistance on the susceptibility of *A. baumannii* to other antibiotics and explored the genetic basis of resistance acquisition.

This study was presented in a poster in ASM microbe 2018, General meeting of American Society of Microbiology 1–22 June 2018 [22].

## 2. Results

### 2.1. Testing the Antimicrobial Susceptibility of Identified A. baumannii Isolates

A total of 110 *Acinetobacter* isolates recovered from different hospitals were identified phenotypically to a level of 98% (108/110). PCR analysis was carried out on all isolates, and a product of the correct size of the *blaOXA-51* gene was detected in 86.36 % (95/110) of the strains, demonstrating that this gene is highly conserved in *A. baumannii*. 

The highest resistance rates, within the 95 *A. baumannii* isolates, were recorded against cefepime 94% and ceftazidime 93%. (Table 1). It is worth mentioning that it is recommended to determine the susceptibility of colistin by measuring its minimum inhibitory concentration (MIC) using broth dilution method. By applying this method, the resistance of colistin regarded only 5% of the total isolates. The *Clinical and Laboratory Standards Institute* CLSI has selected a MIC ≤2 µg/mL as indicating susceptibility and a MIC of ≥4 µg/mL as indicating resistance to colistin. The MIC was determined for 11 *A. baumannii* isolates, as summarized in Appendix A. We compared the results of the broth microdilution method of the eleven strains with the corresponding results of the disc diffusion method and confirmed that the disc diffusion is not reliable to asses colistin resistance in isolates.

By investigating the level of potential cross-resistance among colistin-resistant isolates, we found cross-resistance to the majority of the tested antibiotics, except for tigecycline, which was active against five isolates (Table 1).

### 2.2. Colistin Resistance Induction

Three colistin-resistant variants, designated as MS48Col-R, MS50Col-R, and MS64Col-R, displayed an increased colistin MIC from 0.06, <0.06, and 0.125 to 14, 16, 32 mg/L, respectively, with a six- to nine-fold change (Appendix A). Depending on the MIC values, our induced isolates were considered heteroresistant variants. We have shown that these strains can adapt to colistin. For the strains tested, the concept of heteroresistance may be strictly applied because a resistant *A. baumannii* sub-population kept colistin resistance in antibiotic-free medium. 

The resistance phenotype of the colistin-resistant mutant, derived from clinical strains exposed to colistin pressure, was converted into a stable mutant phenotype after several successive passages in colistin-free medium, indicating that this resistance was irreversible and rapidly gained. 

### 2.3. Susceptibility Profile and Colistin Resistance Acquisition

Colistin resistance acquisition showed no consequences on the antibiotic dynamics of the multi-drug resistance (MDR) strain (MS48Col-R mutant), while a great change was found in the resistance profiles of the originally sensitive strains (MS50Col-R and MS64Col-R mutant) for all antibiotics, except for imipenem and ceftazidime that showed the same profile in the case of MS50Col-R (Table 2).

### 2.4. Colistin Resistance Acquisition and Cell Morphology

Cell morphology was investigated to examine possible alterations induced by colistin resistance acquisition in Col-R mutant strains in comparison with clinical Col-S and Col-R strains (Figure 1). TEM of sensitive strains (Col-S) showed an intact, uniform, and thick cell membrane structure, while Col-R mutant strains had a very thin cell membrane compared to the sensitive strains. This can be due to LPS alteration, accompanied by the loss of affinity towards colistin.

### 2.5. Mutation Analysis of PmrB and LpxACD Based on Nucleotide Sequences 

We investigated the nucleotide sequences of *pmrB*, the sensor kinase of the two-component system, and lipid A biosynthesis genes (*lpxA*, *IpxD*, and *IpxC*). This was done to detect mutations that accounted for the changes in the susceptibility pattern and the cell membrane of our induced mutants by comparing the nucleotide sequences to those of the clinically isolated colistin-resistant (Col-R) and -susceptible (Col-S) strains. 

Based on the sequencing of all genes that were investigated, all strains exhibited 100% DNA sequence similarity in the regions of the *pmrB* and *lpx* genes. The investigation of the gene sequence of *lpxA* and *lpxD*, which encode acyl-transferases involved in lipid A biosynthesis, as well as of *lpxC*, that is involved in lipid A biosynthesis, located in the chromosomal region of *A. baumannii* genome, showed a single replacement in *LpxD* in the MS64Col-R mutant as well as in the resistant clinical isolate (MS34Col-R) corresponding to K117E in the lipid-binding domain (LBD) compared to the susceptible parent strain. This point mutation caused a charge shift in the amino acid from the acidic state of the carboxyl residue to the basic hydrophobic state, leading to polymerization disruption. (Figure 2).

This was confirmed by the predicted 3D structure of the mutated lpxD of the MS64Col-R strain. The 3D structure revealed the impact of the mutation at position 117 in the structure of the lipid-binding domain region of the protein. Besides, resistant and susceptible strains shared five single nucleotide polymorphism (SNP) in the *lpxD* gene that were synonymous upon protein translation.

On the other hand, according to the coding sequences of the *lpxA* and *lpxC* genes, 14 polymorphic positions were noticed that were all synonymous at the protein level (not shown) in the laboratory-induced strain (MS64Col-R) and the resistant clinical strain (MS34Col-R). 

Besides the *lpx* genes, in this report, no mutations were observed within the *pmrB* gene in MS64Col-R mutant. Meanwhile, non-synonymous mutations in *PmrB* were observed in MS34Col-R, corresponding to substitutions and frameshift (deletion) mutations. 

Two amino acid substitutions, A95T and P157R, were detected in the transmembrane domains (TM). A95T was detected in TM1, while P157R was detected in TM2. Here, we suggest that modifications in LPS, not its complete loss, induces low colistin resistance in mutants and clinical *A. baumannii* isolates and can be linked to a single mutation in one *(lpxD)* of the three genes of the lipid A biosynthesis pathway (*lpxA*, *lpxC*, and *lpxD*). Regarding the frameshift mutation, a nonsense mutation was created at position 355 in the ATP-binding domain of the sensor kinase, which, as we suggested, led to premature termination of the amino acid sequence at a critical site, resulting in Δ355 protein (Table 3).

## 3. Discussion

*A. baumannii* is now regarded as one of the most difficult nosocomial-acquired pathogens to treat and control, characterized by a worryingly increasing rate of colistin resistance, observed worldwide [7,23,24]. In our study, we propose that the incidence of MDR *A. baumannii* is very high in Egypt. However, Abdulzahra and his colleagues demonstrated that only two *A. baumannii* isolates were colistin-resistant [24].

Carbepenems have a pivotal role in the treatment of *Acinetobacter*-related infections. There is a worldwide concern regarding the increasing carbapenem resistance in *A. baumannii.* This emerging pattern drastically limits the range of therapeutic alternatives. The most reported carbapenem resistance mechanisms involve the acquisition of carbapenem-hydrolyzing D beta-lactamases. These enzymes are referred to as OXA-23, OXA-24, and OXA-58. In addition to beta-lactamases, carbapenem resistance in *A. baumannii* may also result from penicillin-binding proteins or porin modifications. Several porins, including the 33 kDa CarO protein, might be involved in carbapenem resistance [25]. 

In our study, 70% of the isolates were resistant to imipenem, similar to what reported in a study conducted in 2014 [26]. Unfortunately, imipenem resistance in Egypt has recently increased to around 92% of the studied cases [27,28], indicating the decrease of carbepenem effectiveness against *A. baumannii* infections.

It is known that colistin has poor agar diffusion, which consequently limits the predictive accuracy of the disk diffusion test. Therefore, we applied the microdilution method to determine colistin sensitivity. In fact, the resistance pattern of *A. baumannii* strains differs depending on the assessment method, in this case, the agar and broth microdilution methods. 

According to the definition of heteroresistance, our induced strains were considered as heteroresistant variants [8,9]. A similar phenotype has also been described in clinical isolates of *Stenotrophomonas maltophilia* [29] and carbapenem-resistant *A. baumannii* [30]. Curiously, this phenomenon has enhanced colistin resistance in *A. baumannii* [31], an issue that is now of great interest, since colistin is the drug of choice for *A. baumannii* therapy [32,33,34] 

To answer the question of whether this phenotypic manifestation has a genetic basis, further investigation is required, including genomic sequencing. Choi and Ko stated that the two-component gene system *PmrA/B* is associated with the emergence of colistin heteroresistance in *A.baumannii* [35]. Moreover, although colistin resistance in *A. baumannii* is uncommon, it can be rapidly induced in the lab, as confirmed in our investigation. Choi and Ko proved that colistin resistance can be readily induced during drug therapy by a single-step mutation in *A. baumannii*, *Pseudomonas aeruginosa*, and *Klebsiella pneumoniae* [35].

Moreover, several researchers showed increased incidence of heteroresistant *A. baumannii* isolates, e.g., Ezadi et al., who characterized 20.45% of isolates as colistin-heteroresistant [32], Li et al., who found 93.8% resistance in 16 clinical isolates [1], Hawley et al., who showed 100% resistance in 19 isolates [33], and Charretier and co-workers, who described half of their *A. baumannii* isolates as heteroresistant [36]. This makes it necessary to evaluate the appearance of heteroresistance in *A. baumannii* during colistin treatment.

This investigation connects the acquisition of colistin resistance in clinical *A. baumannii* isolates to variations of the susceptibility to other antibiotics, which can be explained by multiresistance mechanisms active in this species [37], as reported by López-Rojas et al., who linked the acquisition of resistance to colistin in *A. baumannii* to insusceptibility to cefepime and sulbactam [21]. Moreover, Moffat et al. denoted an increase in antibiotic sensitivity as a consequence of colistin resistance acquisition [10]. 

On the other hand, a tendency to overweigh the development of cross-resistance to colistin and antibiotics of different classes has been observed in the MS50Col-R mutant, for which no significant antibiogram alterations occurred between the parent MDR colistin-susceptible strain and its mutated colistin-resistant one, as shown for imipenem and ceftazidime [1,19,38]. Cross-resistance can be explained by the fact that the outer membrane of colistin-resistant strains becomes more impermeable, which increases the resistance to cell wall-targeted antibiotics. Such antibiogram changes lead to a drastic decrease in therapeutic options. 

Another study correlated the generation of cross-resistance to bacitracin to the extensive use of colistin in animal feed [11]. Napier and his colleagues showed the use of colistin was accompanied by cross-resistance to host antimicrobials LL-37 and lysozyme [7]. We suppose that resistance to other antibiotics in *A. baumannii* isolates upon induction of colistin resistance may be related to other antibiotic resistance mechanisms [38].

The variation between our induced colistin-resistant strains could be explained by the presence of LPS mutations. Colistin resistance acquisition resulted in the alteration of the antibiotic resistance profiles. This may be attributed to gene mutations responsible for the displayed resistance, leading to variations in the permeability of the outer membrane [39]. This was suggested by the changes observed in the cell morphology due to alteration of LPS, as confirmed by our investigation and other studies that linked the inactivation of *LpxD* to colistin resistance [40]., 

Knowledge concerning colistin resistance mechanisms in *A. baumannii* isolates of clinical origin remains limited. Recent studies prompted by the discovery of increasing colistin resistance incidence in *A. baumannii* tried to link the observed phenotypes to mutated genes. 

A point mutation (K117P) in the *lpxD* gene matched with mutations detected previously by Nurtop et al. [41], who showed a synergistic interaction between an *IpxD* mutation and mutations in *pmrCAB*, also proved in our clinical colistin-resistant MS34Col-R strain, although this *IpxD* mutation in the induced colistin-resistant *A. baumannii* strain MS64Col-R was the only mutation identified, nevertheless strongly associated with colistin resistance. This point mutation leads to a disruption of polymerization that affects the resulting enzyme binding affinity and consequently causes a modification in LPS [10,42]. Furthermore, a genetic disruption in one of the initiating LPS biosynthesis genes *lpxA*, *lpxC*, or *lpxD* leads to a complete loss of LPS as a result of the lack of lipid A production and, correspondingly, to colistin resistance by decreasing the binding potentials of colistin to the bacterial membrane [10,18]; however, this mechanism leading to colistin resistance is less common of the first one described.

The most common colistin resistance mechanism involves the two-component signal transduction system (PmrA/B) that regulates the modification of endogenous LPS and is also related to polymyxin resistance in *A. baumannii* isolates [19]. This is due to overexpression of or mutations in its domains, i.e., the sensor kinase (*PmrB*) and the response regulator (*PmrA*) [14,43]. 

Therefore, *PmrB* is an important mutational target in the evolution of colistin resistance and appears to be the most commonly mutated gene in *A. baumannii*. Owing to this fact, its gene sequence was investigated in the clinically resistant strains and mutant ones. Our investigation detected non-synonymous mutations in the transmembrane domains of the clinically resistant MS34Col-R strain. According to the current knowledge of the PmrB protein structure [44], position 95 is located into the amino-terminal protein portion including the cytoplasmic secretion signal (aa 1–110) of the TM1 transmembrane domain, which is a domain putatively involved in the delivery of the protein to the cell membrane [45].

In addition, a non-conservative mutation in the transmembrane TM2 domain (aa 111–219) at position P157R was also detected. To the best of our knowledge, these mutations have not been reported elsewhere. Other various transmembrane domain *PmrB* mutations have been reported in other genera, such as the L14P mutation in *P. aeruginosa* [44] and the L10P mutation of *Escherichia coli* [46]. These mutations are located in the same cytoplasmic secretion signal domain of PmrB. Beceiro et al. proposed that these mutation could activate *PmrB* and modulate the permeability of the outer membrane to colistin, leading to the evolution of colistin resistance [18]. 

Recently, among mutations in the *pmrB* gene previously associated with colistin resistance in *A. baumannii* [18,19], a nonsense mutation that was created at position 355 in the ATP-binding domain of the sensor kinase has been identified. Therefore, we suppose that this mutation could induce the constitutive expression of *pmrA*, leading to the upregulation of the *pmrCAB* operon and subsequent synthesis and addition of phosphor-ethanolamine to hepta-acylated lipid A [18].

## 4. Conclusions

The characterized mutations in *PmrB* and *LpxD* in our investigation may have a role in the modification of LPS structure, which affects colistin binding affinity to the cell membrane and decreases the susceptibility of *A. baumannii* to other unrelated drugs. This observed phenotype related to genetic mutations can be further promoted by site-directed mutation in the *IpxD* gene. Our study, worryingly, postulates that colistin, as a drug of last resort, may rapidly become ineffective against MDR *A. baumannii*, due to its extensive use in health care facilities. The rate of resistance is expected to increase, because colistin is used at sub-MIC concentrations with other drugs to decrease its toxicity and create synergistic effects. Moreover, the acquisition of colistin resistance may also be accompanied by resistance to many unrelated antimicrobial compounds. Therefore, knowledge of the mechanisms of colistin resistance in *A. baumannii* will be informative for the development of strategies to overcome resistance issues in the treatment of this rapidly emerging pathogen.

## 5. Materials and Methods

### 5.1. Bacterial Strains

A total of 110 *Acinetobacter* isolates (that were kindly donated by Dr. Nayra Mohamed, Professor of Microbiology and Immunology, Faculty of Pharmacy, Cairo University, Egypt) were recovered from the bloodstream, urinary tract, respiratory and sputum samples, as well as skin of patients admitted to different intensive care units. Identification to the genus level was done phenotypically on the basis of morphological and biochemical properties through the catalase test, oxidase test, and growth on MacConkey agar, Simmons’ citrate agar, blood agar media, and at 44 °C; it was then confirmed by *bla_OXA-51_*-like carbapenemase gene detection, which is intrinsic to this species [47], using the sequences of the primers in Table 4. Strains were stored in Brain Heart–Glycerol (Oxoid Ltd., Basingstoke, UK) at −80 °C until use. Quality control assessment was done using *Staphylococcus aureus* ATCC® 25923 *E. coli* ATCC® 2599, and *A. baumannnii* ATCC 19606.

### 5.2. Antimicrobial Susceptibility Testing

Antibiotic susceptibility testing was performed using the Bauer–Kirby disc diffusion method by applying 10 different types of commercial antibiotic discs (Oxoid™ United Kingdom and stored at −20: +8 °C). These antibiotics were as follows: imipenem (10 µg/disc), amikacin (30 µg/disc), ciprofloxacin (5 µg/disc), ampicillin/sulbactam (10/10 µg/disc), piperacillin/tazobactam (100/10 µg/disc), ceftazidime (30 µg/disc), cefepime (30 µg/disc), colistin sulphate (10 µg/disc), trimethoprim/sulfamethoxazole (1.25/23.75 µg/disc), and tigecycline (15 µg/disc). Breakpoint criteria were established according to the Clinical and Laboratory Standards Institute (CLSI) guidelines [48]. Tigecycline breakpoints were according to Enterobacteriaceae (≤2/≥8 µg/mL for susceptible/resistant) Food and Drug Administration FDA disk diffusion, at >19 mm and ≤14 mm. Colistin discs were used to screen the resistance of *A. baumannii* isolates that showed no inhibition zone. Multi-drug resistance (MDR) isolates were determined based on the resistance to three or more antibiotics classes. MICs of colistin sulphate (Sigma-Aldrich, Inc.) were determined for 11 *A. baumannii* strains (1.5 × 10^5^ CFU/mL) by the broth microdilution method in cation-adjusted Muller Hinton Broth (CA-MHB) (Oxoid, Columbia, MD, USA), according to Clinical and Laboratory Standards Institute methods [48].

### 5.3. Colistin Resistance Induction

Colistin resistance inductions were carried out on susceptible clinical strains, i.e., MS48Col-S (resistant to all tested antibiotics and sensitive to colistin), MS50Col-S, and MS64Col-S (susceptible to almost all tested antibiotics including colistin), through five daily successive passages in increasing colistin concentrations (double dilution), and was demonstrated by a higher level of resistance for each subculture in CA-MHB broth. Briefly, colonies (approximately 10^9^ colony-forming units (CFU)) of each strain were suspended in 100 µL of sterile phosphate-buffered saline (PBS) (0.9%), and 3 μL of each bacterial suspension was inoculated on CA-MHA plates supplemented with increasing concentrations (1–1024 mgl^−1^) of colistin sulphate on consecutive days, until the highest MIC of colistin was reached; the plates were incubated overnight at 37 °C. Colonies (>20 colonies) were taken from the plate with the highest concentration of colistin sulphate from each culture, resuspended in 100 µL of 0.9% PBS, and used to inoculate plates containing colistin. This procedure was repeated for the duration of the experiment (5/6 days) until the defined endpoint was achieved. Daily passages in colistin-free CA-MHB continued for 16 days, for all isolates, to ensure the stability of the mutation. Colonies of each mutant strain were inoculated in Brain Heart–Glycerol with appropriate concentrations of colistin sulphate for storage at −80 °C until checking their identity by the presence of the *bla*_OXA51_ gene, using conventional PCR [49]. 

Antibiotic susceptibilities were retested for the strains subjected to colistin resistance acquisition and compared to those of their parent sensitive isolates subjected to the same sub-culturing conditions, except for the addition of colistin to their corresponding media. 

### 5.4. Cell Morphological Examination Using Thin-Section Transmission Electron Microscopy (TEM)

In our study, we used TEM to compare morphological changes in the cell membranes of clinical colistin-sensitive (MS48Col-S, MS64Col-S), clinical colistin-resistant (MS34Col-R), and induced colistin-resistant (MS48Col-R, MS64Col-R) strains. The five *A. baumannii* strains were sub-cultured in Nutrient broth (NB) (LAB M, USA) for 24 h. Bacterial cells were centrifuged (4000 rpm/10 min), washed with distilled water, and immersed in a freshly prepared 1% (w/v) aqueous potassium permanganate solution for 5 min at room temperature for fixation. Dehydrated specimens were embedded in epoxy resin. Suspensions were submitted to the Regional Center for Mycology and Biotechnology (RCMB), Al-Azhar University, Cairo, Egypt, for agarose embedding and thin-section preparation. Sections were placed on carbon-coated grids and viewed in a JEM-1010, 120-kV transmission electron microscope (JEOL, Peabody, MA, USA) [50]

### 5.5. Nucleotide Sequencing of pmrB and lpx lipid A Biosynthesis Genes

The two-component system sensor kinase gene, *pmrB*, and lipid A biosynthesis genes, *lpxA*, *lpxC*, and *lpxD* was amplified from *A. baumannii* MS34Col-R, MS64Col-S, and MS64Col-R mutant genomic DNA by PCR, using primers (Table 4). The nucleotide sequences of the amplified fragments were determined using ABI 3730xl DNA sequencer (Applied Biosystems, Cologne, Germany). The obtained sequences were analyzed by www.bioinformatics.org and DNA Baser software as sequence assembler and finally confirmed by CLC workbench (www.qiagenbioinformatics.com). BLASTN program was used to search for homologous sequences (www.ncbi.nlm.nih.gov/BLAST/). The reference sequence of *A. baumannii* ATCC 19606 (GenBank accession number HM149345.1) was compared with the sequences of our strains. Protein translation and alignment were done using the Expasy portal at the Swiss Institute of Bioinformatics website (http://web/expasy.org/translate/) as well as the CLUSTALW2 multiple sequence alignment program (http://www.ebi.ac.uk/Tools/msa/clustalw2/) BLASTp and Clustal W for detection of amino acid alterations.

### 5.6. Nucleotide Sequence Accession Numbers

The nucleotide sequences presented in our study for *A. baumannii* strains MS64Col-S, MS64Col-R mutant, and MS34Col-R were submitted to the GenBank database, as follows: *LpxA*: MK024803, MK024804, and MK024805, *LpxD*: MK034358, MK086036, and MK086037, *LpxC*: MK034355, MK034356, and MK034357, *PmrB*1: MK108027, MK108028, and MK108029, *PmrB*2: MK166024, MK166025, and MK166026, respectively.

### 5.7. Protein 3D Model Prediction

Prediction of the 3D structure of the LxpD R mutant protein of MS64Col, resulting from induction of colistin resistance, was performed using SWISS-MODEL tool (https://swissmodel.expasy.org/). 

## Abbreviation

Col-RColistin-resistantCol-SColistin-sensitiveTEMTransmission Electron MicroscopyCA-MHBCation adjustment Muller–Hinton BrothCFUColony-Forming Unit

## Figures and Tables

**Figure 1 antibiotics-09-00164-f001:**
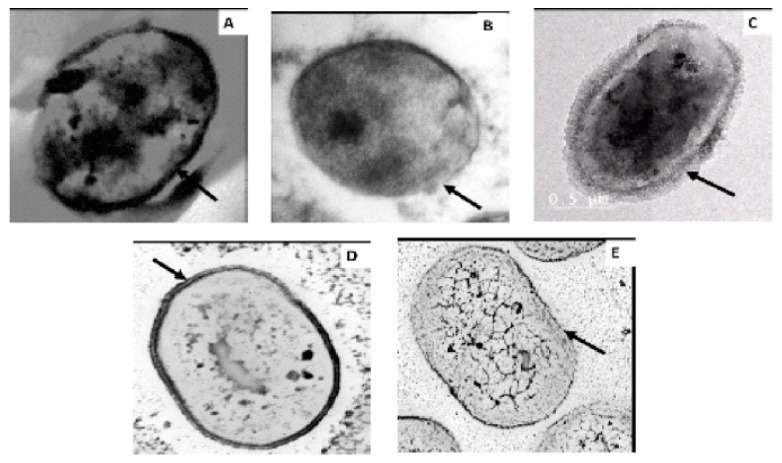
Transmission electron micrographs of thin sections of colistin-sensitive and -resistant *A. baumannii* strains. (**A**) Colistin-sensitive strain MS48, displaying a thick cell membrane. (**B**) MS48 after the acquisition of colistin resistance, displaying a thin cell membrane. (**C**) Clinical colistin-resistant strain MS34, displaying a thin cell membrane and a glycocalyx. (**D**) Clinical colistin-sensitive strain MS64, displaying a thick cell membrane. (**E**) MS64 after the acquisition of colistin resistance, displaying a thin cell membrane.

**Figure 2 antibiotics-09-00164-f002:**
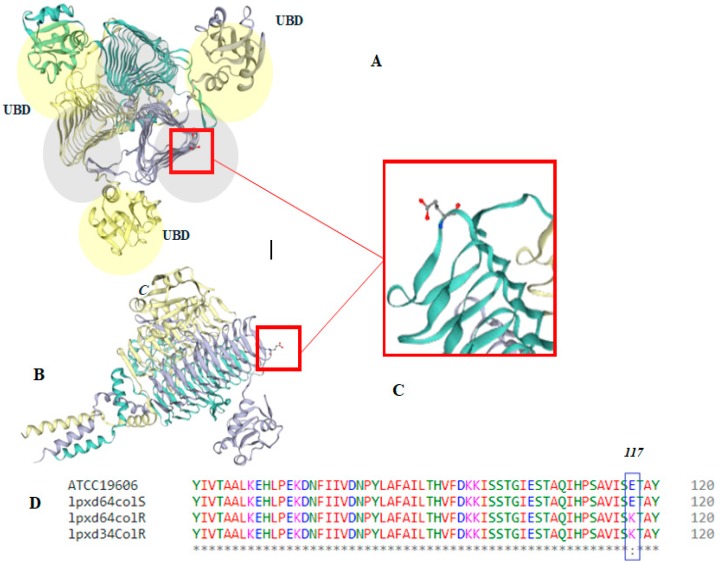
Predicted 3D model structure of MS64Col-R LpxD protein. (**A**) Orthogonal view of the trimer, showing three chains where the lipid-binding domain (LBD) is colored in grey shades. (**B**) Side view of the trimer. (**C**) Amino acid substitution position (117). (**D**) Amino acid sequence alignments of a part of the LpxD protein of the colistin-sensitive reference strain (ATCC19606), colistin-sensitive clinical isolate (64colR), colistin-resistant laboratory-induced isolate (64colR), and colistin-resistant clinical isolate (34colR), generated by online Clustal W–Model server. UBD: uridine-binding domain.

**Table 1 antibiotics-09-00164-t001:** Sensitivity index of *Acinetobacter baumannii* clinical isolates and co-resistance to antibiotics and colistin.

Antibiotic Name		Resistance Level in Clinical *A. baumannii* Isolates						Level of Co-Resistance to Antibiotic and Colistin *	
		R		I		S		No.	%
		No.	%	No.	%	No.	%		
Ampicillin/sulbactam		85	90	3	3	7	7	9	100
Piperacillin/tazobactam		66	70	23	24	6	6	8	88.8
3rd generation	Ceftazidime	88	93	7	7	0	0	9	100
4th generation	Cefepime	89	94	5	5	1	1	9	100
Amikacin		68	72	19	20	8	8	7	77.7
Ciprofloxacin		78	82	12	13	5	5	8	88.8
Imipenem		66	70	2	2	27	28	8	88.8
Tigecycline		21	22	35	37	39	41	4	44.4
Trimethoprim/ Sulfamethoxazole		81	85	6	6	8	9	9	100

* Level of co-resistance was with respect to nine colistin-resistant *A. baumannii* isolates that showed no inhibition zone around the disc.

**Table 2 antibiotics-09-00164-t002:** Antibiotic susceptibility redetermination after colistin resistance induction in comparison to susceptibility before induction.

Clinical *A. baumannii* Isolates Antibiotics	Resistance Pattern					
	MS48D	MS50	MS64	MS48D	MS050	MS64
	Col-S Before Induction			Col-R After Induction		
Colistin MIC (μg/mL)	0.125	<0.06	0.06	14	16	32
CT	S	S	S	R	R	R
TZP	R	S	S	R	R	R
FEP	R	I	S	R	R	R
CIP	R	S	S	R	R	R
IMP	S	S	S	S	S	I
TGC	I	S	S	I	I	I
CAZ	R	R	S	R	R	R
SAM	R	S	S	R	R	R
SXT	R	S	S	R	R	R
AK	R	S	S	R	R	R

CT: colistin, TZP: Piperacillin/tazobactam, FEP: Cefepime, CIP: Ciprofloxacin, IMP: Imipenem, TGC: Tigecycline, CAZ: Ceftazidime, SAM: Ampicillin/sulbactam, AK: Amikacin, SXT: Trimethoprim/Sulfamethoxazole.

**Table 3 antibiotics-09-00164-t003:** Amino acid changes or mutations in *pmrB* and *lpxD* in colistin-resistant clinical and laboratory-induced strains (compared to their parental wild-type strain ATCC 19606).

*Strains*	Colistin (Col) ^a^ MIC (g/mL)	Amino Acid Change(s) in ^b^:				
		*pmrB* (444 aa)				
		*lpxD* (aa356)	TM1 (aa 10–29)	TM2 (142–164)	HisKA (aa 218–280)	HATPaseC (aa 326–437)
MS64Col-S	32					
MS64Col-R	14	K117E				
MS34Col-R	16	K117E	A95T	P157R		355 frameshift

^a^ Colistin (Col) MICs were determined by the broth microdilution method according to the CLSI. ^b^ The predicted domains according to the NCBI domain predictor (www.ncbi.nlm.nhi.gov/protein) are indicated as follows: TM1, TM2, first and second transmembrane domains, HisK, histidine kinase (dimerization/phosphoacceptor) domain, and HATPaseC, histidine-kinase-like ATPase. Only domains or regions displaying mutations or variants are shown. The amino acid (aa) positions corresponding to these domains are displayed in brackets.

**Table 4 antibiotics-09-00164-t004:** Primers used in this study.

Gene Name	Use	Sequence (5′-3′)		bp	Reference
***bla_OXA-51-like_***	*A. baumannii* Identification	F	5′TAATGCTTTGAT CGGCCTTG3′	353	[47]
		R	5′TGGATTGCACTTCATCTTGG3′		
*lpxA*	Lipid A synthesis	F	5′TGAAGCATTA GCTCAAGTTT3′	1181	[10]
		R	5′GTCAGCAAATCAATACAAGA3′		
*lpxD*		F	5′CAAAGTATGAATACAACTTTTGAG3′	1164	
		R	5′GTCAATGGCACATCTGCTAAT3′		
*lpxC*		F	5′TGAAGATGACGTTCCTGCAA3′	1502	
		R	5′TGGTGAAAATCAGGCAATGA3′		
*pmrB1*	Two-Component System	F	5′GTGCATTATTCATTAAAAAAAC 3′	1335	[18]
		R	5′TCACGCTCTTGTTTCATGTA 3′		
*pmrB2*		F	5′GGTTCGTGAAGCTTTCG 3′	599	
		R	5′CCTAAATCGATTTCTTTTTG 3′		

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
