# Peer review of "Acquisition of Colistin Resistance Links Cell Membrane Thickness Alteration with a Point Mutation in the lpxD Gene in Acinetobacter baumannii"

_antibiotics, 2020, doi:10.3390/antibiotics9040164_

Round 1
Reviewer 1 Report
The work presented in the manuscript is very informative and well presented. A.baumannii is linked to nosocomial infection . Here the author have highlighted that genetic aspect like mutations in PmrB and LpxD plays a crucial role altering the LPS of A.baumannii that leads to resistance to colistin and other related drugs. This finding will help in future work focusing on colistinn resistance in A.baumannii .
Minor comments:
1) Please check the punctuations
2) Line 195-197;200-204;210-213: Needs reconstruction.
Author Response
Response to Reviewer 1 Comments on (Acquisition of colistin resistance links cell membrane thickness alteration with a point mutation in lpxD gene in Acinetobacter baumannii)
antibiotics-750849
Point 1: correct the punctuation
Response: line 43-44: Unfortunately,
Line 48: A.baumannii, which latter,aid contributes to in therapeutics implications.
Line 108: (MS48Col-R mutant), while a great change in resistance profiles of Colistin resistance
Line 134 pmrB,(add comma)
Line 144 K117E in the lipid-binding domain (LBD), site compared to the susceptible parent strain. (remove comma)
Line 171: through, substations and frameshift (deletion) (remove comma)
Line 173. 186, 287: Trans-membrane
Line 220: as confirmed in our investigation. Choi and Ko proved that colistin resistance can be readily induced (remove comma)
line 255: add comma
line 288: strain, that remove comma
Point 2: line 195-197;200-204;210-213: Needs reconstruction
Line 195-197: We have been linked to the high-level colistin resistance in A. baumannii to antimicrobial profile, the strains become more resistance and that was proved by López-Rojas et al, who connect the acquisition of resistance to colistin in A. baumannii to susceptibility to cefepime and sulbactam.
Response 2:
line 229-233: This investigation connects the acquisition of colistin resistance in clinical A. baumannii to susceptibility profile of other antibiotics in which strains become more resistant. This was also reported by López-Rojas et al who connect the acquisition of colistin resistance in A. baumannii to cefepime and sulbactam insusceptibility [22].
Line 200-204: On the other side, the overweighed the tendency of cross-resistance between colistin and other antibiotics of different classes. Have been observed for MS 50Col-R mutant that comes to true with study supported by Li and his co-worker, where no significant antibiogram alterations occurred between the parent MDR colistin susceptible strain and its mutated colistin-resistant one. [2].
Response:
Line 237-243:On the other hand, the overweighed tendency of cross-resistance between colistin and other antibiotics of different classes have been observed for MS 50 Col-R mutant, where no significant antibiogram alterations occurred between the parent MDR colistin susceptible strain and its mutated colistin-resistant one as showed in imipenem and ceftazidime and this was supported by Li et al and Adams et al. [22].
Line 210-213: In addition, as a result of that, the acquisition of colistin resistance has been affected the antimicrobial susceptibility profile, we proposed that it could be also affected cell morphology by alteration of LPS as confirmed in our investigation and other studies that linked to the inactivation of LpxD with colistin resistance [29].
Response
Line 255-259:
The variation between our induced colistin strains could be concluded from LPS mutations. Colistin resistance acquisition resulted in alteration of the antibiotic resistance profiles. This may be attributed to the kind of the mutated gene responsible for the displayed resistance and consequently due to variations in the permeability of the outer membrane. [28] this was proposed by the affected cell morphology due to alteration of LPS as confirmed in our investigation and other studies that linked the inactivation of LpxD to colistin resistance [29]
Reviewer 2 Report
Antibiotics-Article revision: “Acquisition of colistin resistance links cell membrane thickness alteration with a point mutation in lpxD gene in Acinetobacter baumannii”
The aim of the study was to explore the impact of colistin resistance on the susceptibility to other antibiotics, also linking the resistance acquisition to its genetic basis.
General observations
The article highlights findings on the resistance to colistin in A. baumannii clinical isolates. In particular, 89/95 isolates were MDR, with 5 isolates resistant to colistin. As a rare occurrence, some isolates were susceptible to imipenem. In 3 colistin-susceptible isolates, resistance to colistin was inducted. Differences in membrane structure composition and in protein sequences of the PmrA/B system were investigated in both colistin-resistant (inducted or not) and colistin-susceptible (parental) strains. Authors stated that substitutions in lpxD and pmrB, here described, were not described so far. Importantly, in 2 cases, inducted resistance to colistin improved resistance traits against other antibiotics (except imipenem). This study could represent a description of Acinetobacter population in a specific geographic area, highlighting mechanisms of resistance to colistin with its genetic basis, and the relative implications during treatment of Acinetobacter-related infections, especially when colistin is used in sub-MIC concentrations.
However, in my opinion, there are some major points to solve in order to improve the quality of the manuscript.
Major criticisms
- Susceptibility for colistin should be assessed by broth microdilution method for all strains. Basing on both CLSI or EUCAST criteria, the susceptibility profile for colistin cannot be investigated by disc diffusion method. There aren’t breakpoint criteria for disc diffusion. Broth microdilution method is the gold standard and the only validated method.
- There aren’t CLSI breakpoint criteria for tigecycline in Acinetobacter.
- Please, indicate phenotypic method used for bacterial identification in M&M.
- Please, specify criteria for MDR definition in M&M.
- Substitution K117E in lpxD (Table 3) has been previously described (Nurtop et al., 2019-MDR). Discuss your data in comparison with this previous report.
- Discussion section: Explain more in detail, and with more references, how inducted resistance to colistin can improve resistance traits to other antibiotics.
- I think that one of the most interesting data is the heteroresistance to colistin. Explain this aspect more in detail and with more citations.
- 70% of isolates were resistant to imipenem. Resistance mechanisms (carbapenemases) should be characterized, having carbapenems a pivotal role in the treatment of Acinetobacter-related infections. Another interesting point was that some isolates were susceptible to imipenem. This is a very rare occurrence that should be discussed.
- Authors should indicate the geographic area, the hospital/s and the year/period of isolation of A. baumannii isolates.
- Facultative: It could be very interesting any information about clonality and genetic correlation to pandemic clones.
Minor points
- Results: the correct form is “MIC ≤2 as susceptible”, not ≥2
- Results: Please, remark that investigation was conducted on 95 isolates
- M&M: Please, indicate the commercial source of antibiotic discs
- Adjust “trimethoprim/sulfamethoxazole” in all over the text
- Adjust “Acinetobacter baumannii” in italics in all over the text
- Adjust “MS 0” in “MS50” in Table 2
- Use terms “resistant” and “resistance” in adequate manner in all over the text
- M&M: specify what means “oropharynx secretions”
- Line 194, page 7: Adjust “K. pneumonia” in “K. pneumoniae”
- Sentence line 109, page 3: strains MS64-R become resistant also to ceftazidime. Please, correct
Author Response
Response to Reviewer 2 Comments on manuscript (Acquisition of colistin resistance links cell membrane thickness alteration with a point mutation in lpxD gene in Acinetobacter baumannii)
antibiotics-750849
Major criticisms
Point 1: - Susceptibility for colistin should be assessed by broth microdilution method for all strains. Basing on both CLSI or EUCAST criteria, the susceptibility profile for colistin cannot be investigated by disc diffusion method. There aren’t breakpoint criteria for disc diffusion. Broth microdilution method is the gold standard and the only validated method.
Response 1: we thank to the reviewers but we didn’t assess the colistin resistance by microdilution method for all strains, it was not in our scope how many strains are resistant to colistin but we were investigating the effect of colistin resistance induction on the susceptibility pattern to other antibiotics, therefore, we did a preliminary antibiogram determination by disc diffusion method to select our candidates then we further insured our candidates resistance to colistin by MIC determination as we know that disc diffusion is not reliable. Finally, we compared the results of broth microdilution method of those eleven strains to their disc diffusion results to emphasize that fact.
Point 2: There aren’t CLSI breakpoint criteria for tigecycline in Acinetobacter
Response 2: we agree with reviewers and we followed tigecycline breakpoints of Enterobacteriaceae (≤2/≥8μg/mL for susceptible/resistant) Food and Drug Administration FDA disk diffusion at >19 mm and ≤ 14mm) for 15µg /disk. We know that using these ranges would produce an error of around 23% and this was satisfactory to our scope. Moreover a study pointed out that none of the microdilution method, E-test or disc diffusion method revealed reliable results in determination of susceptibility of A. baumannii to tigecycline and the clinical response should be the reference to those cases (Tas et al., 2013). Another study recommended the use of disk diffusion, Etest and VITEK 2 for testing tigecycline but with caution (Piewngam and Kiratisin, 2014).
Point 3: indicate phenotypic method used for bacterial identification in M&M.
Response 3:
We thank the reviewer and we add the phenotypic udentification in line 323-234 in M7M section as follow: A total of 110 Acinetobacter isolates were recovered from the bloodstream, urinary tract, respiratory and sputum samples as well as skin of patients admitted to different intensive care units. Identification to the genus level was done phenotypically using morphological, and biochemical properties as catalase test, oxidase test, growth on MacConkey agar, simmons citrate agar, and blood agar media and at 44OC then it was confirmed by blaOXA-51- like carbapenemase gene detection, which is intrinsic to this species [1], using the sequences of the primers in Table 4. Strains were stored in Brain Heart-Glycerol (Oxoid Ltd., Basingstoke, UK) at -80° C until use. Quality control assessment was done using Staphylococcus aureus ATCC® 25923, E.coli ATCC® 2599, and A. baumannnii ATCC 19606
Point 4: specify criteria for MDR definition in M&M.
Response 4: we thank the reviewer and we add in line 342-343 in M&M section as follow: Multi-drug resistance (MDR) isolates were determined based on the resistance to the three or more antibiotics classes.
Point 5: Substitution K117E in lpxD (Table 3) has been previously described (Nurtop et al., 2019-MDR). Discuss your data in comparison with this previous report.
Response 5: we thank the reviewer for his comment and we add in line 271-274 in discussion section
Point mutation (K117P) in lpxD gene was a matched mutation detected previously by Nurtop et al.[2], who showed synergistic interaction between IpxD mutation along with mutations in pmrCAB as proved in our clinical colistin resistance MS34Col-R strain. Although this IpxD mutation in colistin resistance induced in A. baumannii strain MS64Col-R was alone, but it also corresponds to strict colistin resistance. This point mutation lead to disruption of polymerization that in turn affected the enzyme binding affinity and consequently caused a modification in LPS [3,4].
Point 6: Explain more in detail, and with more references, how inducted resistance to colistin can improve resistance traits to other antibiotics.
Response 6: we appreciated the reviewr comment and we add the new discusion in discusion section in Line 278-284:
On the other hand, the overweighed tendency of cross-resistance between colistin and other antibiotics of different classes have been observed for MS50Col-R mutant, where no significant antibiogram alterations occurred between the parent MDR colistin susceptible strain and its mutated colistin-resistant one as showed in imipenem and ceftazidime and this was supported previously [19,20]. This can be explained by that outer membrane of colistin resistant strains became more permeable that increased the susceptibility to cell wall-targeted antibiotics. Such antibiogram changes led to a drastic decrease in the therapeutic options.
line 290-295: Other study colerated the generation of the cross-resistance to bacitracin to the extensive use of colistin in animal feed[14] alonge with Napier and his colleges, who showed that using of colistin was accombaned by cross-resistance to host antimicrobials LL-37 and lysozyme [15].
Point 7: I think that one of the most interesting data is the heteroresistance to colistin. Explain this aspect more in detail and with more citations.
Response 7: we agree and thank the reviewer and we explain the details for heteroresistance in line 188-190 and line 198-202 in discussion section.
Line 188-190: Curiously, this phenomenon have brighten the rapidly increasing level of colistin resistance in A. baumannii [9], however it is now of a great interest, since colistin is the drug of choice for A. baumannii therapy [10,11].
Line 198-202 Moreover, several studies showed the increasing of heteroresistant A. baumannii isolates incidence as represented by Ezadi et al., who characterized 20.45% of isolates as colistin-heteroresistant [10], Li et al. characterized 93.8% in 16 clinical isolates [13], Hawley et al. showed 100% in 19 isolates [11], besides Charretier and co-workers describe half of the isolates as heteroresistant A. baumannii isolates [14], which led to the necessity of evaluating heteroresistance during colistin treatment in A. baumannii.
Point 8: 70% of isolates were resistant to imipenem. Resistance mechanisms (carbapenemases) should be characterized, having carbapenems a pivotal role in the treatment of Acinetobacter-related infections. Another interesting point was that some isolates were susceptible to imipenem. This is a very rare occurrence that should be discussed.
Response 8: we agree with reviewer and add in discussion section with reference in line 189-209
Carbepenems have a pivotal role in the treatment of Acinetobacter-related infections. There is a worldwide concern because of the increasing trend of carbapenem resistance in A. baumannii.This emerging pattern limits, drastically, the range of therapeutic alternatives. The most reported carbapenem resistance mechanisms are due to the acquisition of carbapenem-hydrolysing class D beta-lactamases. These enzymes are denoted by OXA-23, OXA-24 and OXA-58. In addition to beta-lactamases, carbapenem resistance in A. baumannii may also result from penicillin-binding protein or porin modifications. Several porins, including the 33-kDa CarO protein might be involved in carbapenem resistance (Poirel and Nordmann, 2006).
In our study 70% of the isolates were resistance to imipenem similar to the study conducted in 2014 (Al-Agamy et al., 2014). Unfortunately, imipenem resistance in Egypt has recently increased to around 92% of the studied cases (Al-Hassan et al., 2019; Benmahmod et al., 2019) giving a clue about the deterioration of carbepenem effectiveness against A. baumannii infections.
Point 9: Authors should indicate the geographic area, the hospital/s and the year/period of isolation of A. baumannii isolates.
Response 9: we thank reviewer but the isolates were generously donated by Dr, Nayra Mohamed, professor of Microbiology and Immunology, Faculty of Pharmacy, Cairo University, Egypt).
Point 10; Facultative: It could be very interesting any information about clonality and genetic correlation to pandemic clones.
Response: we thank the reviewer but we didn’t carry out any clonal diversity studies using RAPID or ERIC PCR because it was not in our scope
Minor points
- Results: the correct form is “MIC ≤2 as susceptible”, not ≥2
Response: line 86: The CLSI has selected a MIC of ≤2 µg/ml as susceptible and a MIC of ≥4 µg/ml as resistant to colistin
- Results: Please, remark that investigation was conducted on 95 isolates
Response: in line 80-82, we conducted it
- M&M: Please, indicate the commercial source of antibiotic discs
Response: line 334-335: Oxoid™ United Kingdom and stored at -20: +8°C.
- Adjust “trimethoprim/sulfamethoxazole” in all over the text
Response: line 338: trimethoprim/sulfamethoxazole adjust in M& M
Adjust “Acinetobacter baumannii” in italics in all over the text
Response: we thank the reviewer and we did it all over the text
Point 3: Adjust “MS 0” in “MS50” in Table 2
Response: we thank the reviewer and we add in Table 2 MS50
Use terms “resistant” and “resistance” in adequate manner in all over the text
Response: we agree with reviewer it need to correct and we did it all over the text (line 100,
- M&M: specify what means “oropharynx secretions”
Response: we agree with the reviewer and in line 321 we changed it to sputum sample
- Line 194, page 7: Adjust “K. pneumonia” in “K. pneumoniae”
Response: we agree with the reviewer Line 323: we adjust K. pneumoniae
- Sentence line 109, page 3: strains MS64-R become resistant also to ceftazidime. Please, correct
Response: line 110-111: ceftazidime that have the same profile without change in case of MS50Col-R (Table 2).
New reference as in response to point 8
Al-Agamy, M. H., Khalaf, N. G., Tawfick, M. M., Shibl, A. M. and El Kholy, A. (2014) Molecular characterization of carbapenem-insensitive Acinetobacter baumannii in Egypt. Int J Infect Dis. 22, 49-54.
Al-Hassan, L., Zafer, M. M. and El-Mahallawy, H. (2019) Multiple sequence types responsible for healthcare-associated Acinetobacter baumannii dissemination in a single centre in Egypt. BMC Infect Dis. 19, 829.
Benmahmod, A. B., Said, H. S. and Ibrahim, R. H. (2019) Prevalence and Mechanisms of Carbapenem Resistance Among Acinetobacter baumannii Clinical Isolates in Egypt. Microb Drug Resist. 25, 480-488.
Poirel, L. and Nordmann, P. (2006) Carbapenem resistance in Acinetobacter baumannii: mechanisms and epidemiology. Clin Microbiol Infect. 12, 826-36.
Reviewer 3 Report
In the manuscript “Acquisition of colistin resistance links cell membrane thickness alteration with a point mutation in lpxD gene in Acinetobacter baumannii” of Neveen M Saleh and colleagues, the authors reported on the occurrence and the genetic basis of colistin resistance in Acinetobacter baumannii. Out of 110 A. baumannii isolates, nine were confirmed to be resistant to colistin. The genetic basis of the isolates was assessed and colistin resistance was backed to chromosomal nucleotide polymorphisms.
This reviewer feels that the provided data were interesting. However, the manuscript seems to be based only on observations that need to be confirmed experimentally. It is the opinion of this reviewer that the prevailing manuscript comprises not enough novel information for publication in Antibiotics. A broader study of the isolates and a higher content of colistin-resistant isolates would improve the dataset and the impact of the provided results.
Line 51, 58, 2,…: please check the typos of all bacterial designations and genes (italics) all over the manuscript.
Table 1: Why are some cells indicated in yellow… please describe if necessary.
It would be also helpful if a complete table with the exact MIC values can be provided in the Supplemental material section
Line 107: What would you like to say precisely (massive antibiotic dynamics!)
Line 111-115: Is this result really significant on the basis of only three isolates… this reviewer fells that it is only a observation and needs thus rephrased. A experiment using an isogenic system with and without mutation should provide a final result…
The Figure should be cited here…
Table 2: MS 0 … please check
What’s the outcome of Table 2 and where is the information summarized and discussed.
5.1 how were the isolates recovered
5.2 which ATCC or NCTC control isolates were used for quality assessment
Author Response
Response to Reviewer 3 Comments on (Acquisition of colistin resistance links cell membrane thickness alteration with a point mutation in lpxD gene in Acinetobacter baumannii)
antibiotics-750849
Point 1: Line 51, 58, 2,…: please check the typos of all bacterial designations and genes (italics) all over the manuscript.
Response: we agree with the reviewer and we did it all over the text , line 3, 26, 31, 51, 58, 62, 64, 67, 69, 70, 74, 80, 135, 140, 141, 142,143, 171, 181, 218, 283, 284, 294, 297, 302, 306,
Point 2: Table 1: Why are some cells indicated in yellow… please describe if necessary.
Response: we agree with the reviewer and it is not significant, just try to describe cross-resistance to colistin, (we removed the yellow color)
Point 3: It would be also helpful if a complete table with the exact MIC values can be provided in the Supplemental material section
Response: we thank the reviewer but we added the required table in supplemental material section (Table S1)
Point 4: Line 107: What would you like to say precisely (massive antibiotic dynamics!)
Response: line 108: we change massive antibiotic dynamics as followagreat change in resistance profiles of the originally sensitive strains
In our investigation we try to describe that there was a great change in resistance profiles of the induced colistin resistant A. baumannii strains through the screening of other antibiotics.
Point 5: Line 111-115: Is this result really significant on the basis of only three isolates… this reviewer fells that it is only an observation and needs thus rephrased. An experiment using an isogenic system with and without mutation should provide a final result…
The Figure should be cited here…
Response:
We thank the reviewer for his comment, but we would like to clarify that all mutations that took place in colistin resistance related genes due to the induction were silent except for the reported K117E in IpxD. This would much resemble the outcome of site directed isogenic mutation system. Moreover, depending on the colistin mechanism in A. baumannii, there is a direct relationship between colistin resistance and cell morphology, accordingly we supposed that three isolates would be enough to ensure the occurrence of these morphological changes.
Figure (1) is cited in the text in line-118
Point 6: Table 2: MS 0 … please check
Response: It has been corrected to MS50
Point 7: What’s the outcome of Table 2 and where is the information summarized and discussed.
Response: we thank the reviewers for his comment and in Table 2 we represent how colistin resistance acquisition affected the resistance profiles of other tested antibiotics.
The results are summarized in this section: Susceptibility Profile and Colistin Resistance Acquisition line 107-111 and discussed in line 237-254.
Point 8: 5.1 how were the isolates recovered
Response: The isolates were kindly donated by Dr Nayra Mohamed, prof. of Microbiology and Immunology, Faculty of Pharmacy, Cairo University, Egypt
Point 9: 5.2 which ATCC or NCTC control isolates were used for quality assessment
Response: We agree with reviewer and we add in line Quality control assessment was done using Staphylococcus aureus ATCC® 25923 and E.coli ATCC® 2599 and A. baumannii ATCC 19606
Round 2
Reviewer 2 Report
Revision 2
Point 1 – If you use disc-diffusion to screen resistance to colistin you cannot say that 10% of 95 isolates were resistant to colistin (because you cannot consider the 90% of susceptible strains with this method).
- Hence you should delete this data from the Results, and you can also refer to the 11 strains that showed “no inhibition zone around the disc” (OK for the Supplementary Table).
- You should also delete data about colistin in Table 1.
- Moreover, you should clarify in M&M that colistin disc was used only to screen isolates that showed “no inhibition zone around the disc”, in order to deeply investigate them.
Point 6 – Line 227, page 8. I think that the more permeable and thin cell outer membrane in colistin-resistant strains “increased resistance” and not “susceptibility” to cell wall-targeted antibiotics (see pipera/tazo, ampicillin/sulbactam, ceftazidime and cefepime).
Author Response
Response to Reviewer 2 Comments: Revision 2
Response to Reviewer 3 Comments on (Acquisition of colistin resistance links cell membrane thickness alteration with a point mutation in lpxD gene in Acinetobacter baumannii)
antibiotics-750849
Point 1: – If you use disc-diffusion to screen resistance to colistin you cannot say that 10% of 95 isolates were resistant to colistin (because you cannot consider the 90% of susceptible strains with this method).
- Hence you should delete this data from the Results, and you can also refer to the 11 strains that showed “no inhibition zone around the disc” (OK for the Supplementary Table).
Response 1: we agree with the reviewer and we deleted in results section in line 83-84 and in Table 1
|
Antibiotic name |
Resistance level in clinical A. baumannii isolates |
Level of Co-resistance with colistin* |
|||||||
|
R |
I |
S |
No. |
% |
|||||
|
No. |
% |
No. |
% |
No. |
% |
||||
|
Ampicillin/sulbactam |
85 |
90 |
3 |
3 |
7 |
7 |
9 |
100 |
|
|
Piperacillin/tazobactam |
66 |
70 |
23 |
24 |
6 |
6 |
8 |
88.8 |
|
|
3rd generation |
Ceftazidime |
88 |
93 |
7 |
7 |
0 |
0 |
9 |
100 |
|
4th generation |
Cefepime |
89 |
94 |
5 |
5 |
1 |
1 |
9 |
100 |
|
Amikacin |
68 |
72 |
19 |
20 |
8 |
8 |
7 |
77.7 |
|
|
Ciprofloxacin |
78 |
82 |
12 |
13 |
5 |
5 |
8 |
88.8 |
|
|
Imipenem |
66 |
70 |
2 |
2 |
27 |
28 |
8 |
88.8 |
|
|
Tigecycline |
21 |
22 |
35 |
37 |
39 |
41 |
4 |
44.4 |
|
|
Trimethoprim/ Sulfamethoxazole |
81 |
85 |
6 |
6 |
8 |
9 |
9 |
100 |
|
*Level of co-resistance was according to 9 colistin resistance A. baumannii isolates that showed no inhibition zone around the disc.
In M& M section we clarify that colistin disc were used to screen the resistance A. baumannii isolates that showed no inhibition zone, meanwhile it is independent method.
Point 6 – Line 227, page 8. I think that the more permeable and thin cell outer membrane in colistin-resistant strains “increased resistance” and not “susceptibility” to cell wall-targeted antibiotics (see pipera/tazo, ampicillin/sulbactam, ceftazidime and cefepime).
Response 6: We agree with the reviewer and we apologize for that inconvenient mistake and we correct it in Line 227
This can be explained by that outer membrane of colistin resistant strains became more impermeable that increased the resistance to cell wall-targeted antibiotics.
Reviewer 3 Report
I thank the authors for the revsions to my specific comments, which were carefully done.
It is still my opinion that the influence of the mutations needs to be confirmed by mutational analysis and knockout eyperiments. However, in the view of both other reviewers, I will not impede the publication of the manuscript only on the basis of my "opinion".
As my specific points were adresse carefully, I will judge that the manuscript is suited for publication.
Author Response
Response to Reviewer 3 Comments: Revision 2
Response to Reviewer 3 Comments on (Acquisition of colistin resistance links cell membrane thickness alteration with a point mutation in lpxD gene in Acinetobacter baumannii)
antibiotics-750849
Point 1: – It is still my opinion that the influence of the mutations needs to be confirmed by mutational analysis and knockout experiments.
Response 1: we respect the reviewer opinion but unfortunately, due to the crisis of corona, all the laboratory is shutdown as well as we didn’t have the financial aid to do knockout experiments
We can add in the conclusion section line 281-282
This observed phenotype related to genetic mutation can be further investigated by site directed mutation in IpxD gene